Myiarchus flycatchers are the primary seed dispersers of Bursera longipes in a Mexican dry forest

Almazán-Núñez R. Carlos 1
Eguiarte Luis E. 2
Arizmendi María del Coro 3
Corcuera Pablo 4 pcmr@xanum.uam.mx
1 Laboratorio Integral de Fauna Silvestre, Unidad Académica de Ciencias Químico Biológicas, Universidad Autónoma de Guerrero , Chilpancingo, Guerrero , Mexico
2 Laboratorio de Evolución Molecular y Experimental, Departamento de Ecología Evolutiva, Instituto de Ecología, Universidad Nacional Autónoma de México , Mexico, DF , Mexico
3 Laboratorio de Ecología, Unidad de Biotecnología y Prototipos, Universidad Nacional Autónoma de México , Mexico, DF , Mexico
4 Departamento de Biología, Universidad Autónoma Metropolitana-Iztapalapa , Mexico, DF , Mexico
Rodriguez-Lanetty Mauricio
Electronic publication date: 2016 Jun 15
Publication date: 2016
Volume: 4
Electronic Location ID: e2126
Received 2015 Nov 12; Accepted 2016 May 23
Copyright: © 2016 Almazán-Núñez et al.
Copyright year: 2016
Copyright holder: Almazán-Núñez et al.
License: This is an open access article distributed under the terms of the Creative Commons Attribution License, which permits unrestricted use, distribution, reproduction and adaptation in any medium and for any purpose provided that it is properly attributed. For attribution, the original author(s), title, publication source (PeerJ) and either DOI or URL of the article must be cited.
License URL: https://creativecommons.org/licenses/by/4.0/

Keywords: Germination, Establishment, Nurse plants, Restoration, Flycatchers, Coevolution, Seedling

Funding: This project was benefited from the financial support of projects DGAPA-PAPIIT No.IN217511 and CONABIO HQ008 IN210908 on behalf of M.C. Arizmendi. The Metropolitan Autonomous University gave financial support and the National Council of Science and Technology (CONACYT; Reg 165552) offered a scholarship to the first author. The funders had no role in study design, data collection and analysis, decision to publish, or preparation of the manuscript.

==============================
We evaluated the seed dispersal of Bursera longipes by birds along a successional gradient of tropical dry forest (TDF) in southwestern Mexico. B. longipes is an endemic tree to the TDF in the Balsas basin. The relative abundance of frugivorous birds, their frequency of visits to B. longipes and the number of removed fruits were recorded at three study sites with different stages of forest succession (early, intermediate and mature) characterized by distinct floristic and structural elements. Flycatchers of the Myiarchus and Tyrannus genera removed the majority of fruits at each site. Overall, visits to B. longipes were less frequent at the early successional site. Birds that function as legitimate dispersers by consuming whole seeds and regurgitating or defecating intact seeds in the process also remove the pseudoaril from seeds, thereby facilitating the germination process. The highest germination percentages were recorded for seeds that passed through the digestive system of two migratory flycatchers: M. cinerascens and M. nutingii. Perch plants, mainly composed of legumes (e.g., Eysenhardtia polystachya, Acacia cochliacantha, Calliandra eryophylla, Mimosa polyantha), serve also as nurse plants since the number of young individuals recruited from B. longipes was higher under these than expected by chance. This study shows that Myiarchus flycatchers are the most efficient seed dispersers of B. longipes across all successional stages. This suggests a close mutualistic relationship derived from adaptive processes and local specializations throughout the distribution of both taxa, as supported by the geographic mosaic theory of coevolution.

Introduction

Biotic seed dispersal plays a central role in the spatial dynamics of plant populations (Spiegel & Nathan, 2007; Schupp, Jordano & Gómez, 2010; Jordano et al., 2010; Forget et al., 2011). Dispersion may encourage establishment of plants, since intraspecific competition is generally lower in sites distant from the parent plant (Godínez-Álvarez, Valiente-Banuet & Rojas-Martínez, 2002; Schupp, Milleron & Russo, 2002). Moreover, the incidence of pathogens and predators is usually lower at new sites, where the seeds are dispersed by animals (Jordano et al., 2010; Obeso, Martínez & García, 2011).

In arid and semi-arid environments, biotic dispersal, germination and seedling establishment are critical phases of the plant life cycle (Valiente-Banuet et al., 1991; Godínez-Álvarez & Valiente-Banuet, 1998; Orozco-Almanza et al., 2003; Padilla & Pugnaire, 2006). During the dry season, for example, seedlings face adverse conditions, such as dry soil, direct sunlight and extreme temperatures (Godínez-Álvarez & Valiente-Banuet, 1998). However, dispersers may deposit seeds in preferable microhabitats (i.e., under nurse plants) that promote germination and seedling survival (Pérez-Villafaña & Valiente-Banuet, 2009). Yet in arid environments, few microhabitats with suitable conditions exist, and certain microhabitats may in fact adversely affect seedling establishment (Calviño-Cancela & Martín-Herrero, 2009). Thus, the role of dispersers in depositing seeds in sites with appropriate conditions for germination in these environments is particularly important because there are few adequate microhabitats in which seeds can be established (Vasconcellos-Neto, Barbosa de Albuquerque & Rodrigues Silva, 2009).

The relative contribution of seed dispersal by birds towards plant fitness has been assessed by “seed dispersal effectiveness,” or SDE (Schupp, 1993; Schupp, Jordano & Gómez, 2010). Seed dispersal effectiveness has a quantitative (i.e., number of seeds dispersed) and a qualitative (i.e., probability that seeds reach adult stage) component that is mainly dependent on feeding behavior and movement patterns of dispersers (Calviño-Cancela, 2002; Calviño-Cancela & Martín-Herrero, 2009), in addition to other functional traits, such as body size (Escribano-Avila et al., 2014). For example, retention time of seeds may vary between different bird species depending on gut length, and as a consequence, dispersers defecate seeds with different degrees of scarification (i.e., the process by which gastric juices weaken the seed coat, encouraging germination; Robertson et al., 2006). Large differences in SDE among dispersers generally signifies that dispersion processes are complementary and non-redundant (Escribano-Avila et al., 2014). However, differences in effectiveness depend not only on morphological traits but also on deposition of seeds in suitable microhabitats (e.g., type of microhabitat; Calviño-Cancela & Martín-Herrero, 2009).

Different aspects of seed dispersal are being modified by the transformation of natural communities worldwide by human activities (Wright, 2007). For instance, inadequate agricultural practices have reduced once undisturbed portions of tropical dry forest (TDF) in Mesoamerica to fragments of various successional stages (Quesada et al., 2009). Since the soil seed bank may be considerably reduced in disturbed fragments (Uhl, 1987; Martins & Engel, 2007), seed dispersal can play an important role in the recruitment of plants and hence contribute towards the composition and density of woody plants and the eventual restoration of these forests (Hammond, 1995; Peña-Claros & De Boo, 2002).

The vegetation of the Balsas basin in southwestern Mexico consists mainly of TDF (Rzedowski, 1978). In these forests, the dominant arboreal elements generally belong to the genus Bursera, or Burseraceae (Rzedowski, Medina Lemos & Calderón de Rzedowski, 2005; Almazán-Núñez et al., 2012), whose fruits are a food source for resident and migratory frugivorous and insectivorous birds during the dry season (Ortiz-Pulido & Rico-Gray, 2006; Ramos-Ordoñez & Arizmendi, 2011). Furthermore, the distribution of these birds throughout different successional stages of TDF is related to the presence of these trees (Almazán-Núñez et al., 2015).

Although the role of birds as consumers and dispersers of Bursera spp., including flycatchers and vireos, has been previously described (Greenberg, Foster & Marquez-Valdelamar, 1995; Hammond, 1995; Ortiz-Pulido & Rico-Gray, 2006; Ramos-Ordoñez & Arizmendi, 2011), there are few detailed reports on this subject. Several examples include the studies of Ramos-Ordoñez & Arizmendi (2011), who performed an analysis of seed dispersal by B. morelensis, and Ortiz-Pulido & Rico-Gray (2006), who examined the same process in B. fagaroides. No additional studies have been performed in TDF to evaluate or compare biotic seed dispersal between sites with different degrees of disturbance (Hammond, 1995; Quesada et al., 2009). Furthermore, the majority of studies on frugivory and seed dispersal have been limited to single locations (Ortiz-Pulido & Rico-Gray, 2006; Ramos-Ordoñez & Arizmendi, 2011). This represents a significant potential bias, as patterns that remain elusive at local scales may emerge in regional studies covering areas with varying levels of disturbance or recovery (Carlo, Aukema & Morales, 2007). Frugivory and seed dispersal may also differ across a geographic mosaic, as interacting species may not necessarily have the same distribution (Thompson, 2005). There is a need for studies to elucidate patterns of biotic seed dispersal and seedling establishment in TDF, considering different levels of perturbation or seral stages, in order to create a scientific basis for the application of management and conservation strategies in these forests.

In this study, we describe the primary seed dispersal of B. longipes by frugivorous birds in a TDF of the Balsas basin of Guerrero. B. longipes belongs to the Simaruba sub-group of the diverse Bursera genus and forms part of a larger clade that emerged during the Miocene slightly over 8.5 million years ago (De-Nova et al., 2012). In addition, it is a dominant tree species (Almazán-Núñez et al., 2012) in this biotic region, which is considered to be the center of diversification for this genus in Mesoamerica (Rzedowski, Medina Lemos & Calderón de Rzedowski, 2005). While B. longipes abundance increases in well-preserved sites, it is also found in disturbed areas (Rzedowski, Medina Lemos & Calderón de Rzedowski, 2005). The adaptability of B. longipes to disturbed environments may promote the regeneration of TDF of the Balsas basin and help to reverse fragmentation (Ramos-Ordoñez, Arizmendi & Márquez-Guzmán, 2012). However, its seeds have a hard coat and must be consumed by a frugivore in order to be scarified, and afterwards dispersed (Ramos-Ordoñez & Arizmendi, 2011).

The study was centered on the following questions: (1) Which bird species remove B. longipes seeds along a successional gradient of TDF? (2) Do seeds that pass through the digestive tract of birds have higher germination rates than those that fall from trees? (3) Do differences exist in the quality of seeds dispersed by birds? (4) Does B. longipes require nurse plants in different successional stages? (5) Do seed-dispersing birds preferentially use nurse plants for perching across all successional stages?

Materials and Methods

Study sites

We conducted the study at three different successional stages of TDF that have been mostly unmanaged for varying periods of time since their last major disturbance (i.e., clear-cutting or burning). The three stages are described as follows: (1) The early successional stage (last disturbed ca. 20 y ago) consisted of vegetation regrowth in a plot once used for cattle ranching and, to a lesser extent, seasonal agriculture; (2) The intermediate successional stage (last disturbed ca. 35 y ago) corresponded with a transitional phase between a mature forest and fragmented areas with a matrix of pasture and seasonal corn and bean fields, yet was once dedicated to seasonal corn production and cattle ranching. Structural and floristic elements had developed that mirrored the original dry forest vegetation, to a large extent; (3) The mature successional stage was characterized by a closed canopy and presence of tree cover typical of mature dry forest (i.e., dominance of the Bursera spp.). This stage has not experienced a large scale disturbance for more than 50 y.

The successional stages were found in patches with areas of 97 ha (early stage), 45.3 ha (intermediate stage) and 24 ha (mature stage). The mean distance between successional stages was ca. 1 km (Fig. 1).

Figure 1 Projection of the (A) Balsas basin, (B) distribution of Bursera longipes in the biotic province of the Balsas basin (C) and study area.

Photograph of B. longipes in the (D) rainy season with presence of foliage, (E) in the dry season with the presence of ripe fruit and (F) pseudoaril overlaying the seeds.

Bursera longipes

The genus Bursera is a distinctive component of TDF in Mesoamerica and is composed of ca. 107 species (De-Nova et al., 2012). Its distribution spans from northern Mexico to the northern region of South America (Becerra et al., 2012). The diversification of this genus is related to the southward expansion of TDF in response to the elevation of the Sierra Madre del Sur and the Mexican Volcanic Belt (De-Nova et al., 2012). Bursera evolutionary history indicates that a large portion of the biological richness of Mesoamerican TDF was derived from accelerated rates of speciation in habitats, from the early Miocene to the Pliocene, during pronounced arid periods (Becerra, 2005; Dick & Wright, 2005). This scenario matches other hypotheses that these lineages were mostly restricted to dry environments of Mexico and evolved during long periods of isolation (Valiente-Banuet et al., 2004).

In particular, Bursera longipes is endemic to TDF in the states of Mexico, Morelos, Puebla, Guerrero and Oaxaca in the Balsas basin (Fig. 1; Rzedowski, Medina Lemos & Calderón de Rzedowski, 2005). It is a deciduous species with trivalvate fruits that turn red at maturity; seeds have a slightly orange pseudoaril (Guízar & Sánchez, 1991). Fruits are 1.3 ± 0.02 cm (mean ± SE) in length and 0.87 ± 0.04 cm in width, with a fresh weight of 0.62 ± 0.01 g (N = 100 fruits). Flowering season begins with the onset of the rainy season (May or June), and fruiting occurs in early June or from May–July. Most fruits ripen between November and May.

Seed dispersal effectiveness

In this study we used the quantitative and qualitative components proposed by Schupp (1993) and Schupp, Jordano & Gómez (2010) to estimate the effectiveness of B. longipes seed dispersal in each successional stage. The quantitative component included the frequency of visits to B. longipes tree and the average number of fruits removed per visit by frugivorous birds. The qualitative component was based on the percentage of germination after seeds passed through the digestive system of birds, the probability of seed deposition under a nurse plant (adult plants that positively influence the recruitment of young seedlings) and the possible contribution of bird species to the establishment of B. longipes after seed deposition under nurse plants (Schupp, Jordano & Gómez, 2010). Seed dispersal effectiveness of each frugivore is calculated as the product of the subcomponents of quantity and quality, according to the following expression (Schupp, 1993; Schupp, Jordano & Gómez, 2010): Effectiveness = frequency of visits × average number of removed fruits per visit × proportion of seed germination × seed deposition probability in potential suitable microhabitats × contribution of birds to site of establishment.

Quantitative component

Frequency of visits and average amount of removed fruit

Frequency of visits was determined by focal observations using binoculars (8 × 40 mm). Observations were focused on seven B. longipes individuals with ripe fruits at each successional stage and performed during January–May 2011 and March–May 2012 in the morning (0700–1130 h) and afternoon (1600–1830 h), when bird activity is higher. Each of the seven trees was sampled in each successional stage during both years. A total of 70 h of observation was recorded for each successional stage (10 h/tree), for a total of 210 h in all three stages.

Each tree was observed at a distance of ∼20–30 m for an observational period of 30 min, during which the frugivore species and number of visits were recorded, in addition to number of individuals, total time of visit (from arrival to departure) and number of fruits consumed per visit. The frequency of visits was analyzed in a χ2 contingency table to determine differences among successional stages. In this case, the null hypothesis would indicate the existence of an equal number of visits between successional stages. The number of removed fruits was compared among stages with an unbalanced one-way ANOVA. For this analysis, data were transformed (log x + 1) to meet assumptions of normality and homogeneity of variance.

Qualitative component

Seed germination

Seeds obtained from the faeces of birds captured by nine 12 m mist nets in each successional stage were used to determine the effect of passage of seeds through bird digestive system on the proportion of germination. Mist nets were placed during the months with greatest availability of mature B. longipes fruits (May and December 2010, January–May and December 2011 and March–May 2012). During each period, sampling was performed for 15 days. Mist nets remained open from 07:00 to 18:00 h, resulting in a total of 1,485 net-hours per stage and 4,455 net-hours for all successional stages.

Captured birds were placed in individual cages (40 × 40 cm) lined with mosquito netting and fed ad libitum with ripe B. longipes fruits for a day after capture. Retention time of seeds was estimated from the moment of fruit consumption until defecation or regurgitation of seeds. Birds placed in cages were monitored every 10 min, with the least possible disturbance, in order to confirm their consumption of seeds with pseudoarils. Defecation times were recorded at 10 min inspection intervals. Retention time of seeds is therefore an approximation, according to these intervals.

The premise in this portion of the study is that longer retention times would likely result in seeds being spread farther from the mother plant (Westcott & Graham, 2000). After evacuation, feces were collected, and birds were released. Since the techniques used to collect regurgitated seeds and feces were non-invasive, special authorizations were not required.

Seed viability was tested via flotation tests, in which floating seeds were considered nonviable due to lack of embryonic development (Thompson, Grime & Mason, 1997). Viable seeds were washed with 10% sodium hypochlorite, planted in cotton on petri dishes at ambient temperature and moistened daily with distilled water. This procedure was performed with seeds obtained from different sources and with different treatments as follows: (a) control group 1: seeds with pseudoaril obtained directly from the trees, (b) control group 2: seeds without pseudoaril obtained from trees and (c) seeds that passed through the digestive system of birds. For the final treatment, the germination experiment was only performed with bird species from which the largest number of seeds was obtained: Myiarchus nuttingi (N = 67), Myiodynastes luteiventris (N = 58), Myiarchus cinerascens (N = 33), Melanerpes chrysogenys (N = 29) and Myiarchus tyrannulus (N = 27). A total of 50 seeds per successional stage were used for each of the controls (fruits obtained from trees).

In the first phase of the germination experiment, we mixed the seeds of the control groups from all three stages and assigned a random number to each control treatment. Each control group had a total of five replicates with 30 seeds each. The objective of this portion of the experiment was to evaluate the effect of deinhibition and scarification (Robertson et al., 2006).

Germination experiments were performed directly in the field, where boxes with B. longipes seeds were placed under the canopy of nurse plants Mimosa polyantha and Senna wislizenni, which are commonly used by disperser birds for perching. The boxes were protected with mesh mosquito netting and boric acid was poured around the perimeter to avoid predation by ants. Over the course of 20 days, boxes were checked daily to count the number of germinated seeds. The emergence of a radicle indicated germination.

The estimated retention time that seeds remained in the digestive system of birds was compared between treatments with a one-way ANOVA, following a prior analysis of normality and homoscedasticity. The null hypothesis was that time of retention would be the same for all treatments. Multiple comparisons were analyzed with a Tukey HSD. The percentage of germinated seeds was analyzed using a generalized linear model (GLM) with a binomial distribution error and a logit link function (Crawley, 2012) to determine significant differences between treatments. Post hoc pairwise t-test comparisons were carried out for each germination treatment. To evaluate the effect of passage time through the bird gut on seeds on germination percentage, we performed a linear regression (Traveset, 1998). In addition, we also analyzed the relationship between the total body length of captured birds and the percentage of germinated seeds. The null hypothesis of this test was that a positive relationship would exist between these variables. We measured body length using a digital calibrator, according to the specifications given by Ralph et al. (1996). Analyses were performed with SPSS 17.0 software (SPSS, Inc., Chicago, IL, USA).

Potential suitable sites for recruitment

Frugivorous birds deposit faeces under their perch trees (Vasconcellos-Neto, Barbosa de Albuquerque & Rodrigues Silva, 2009), but only the canopy of certain shrubs and trees provides suitable conditions for recruitment of seeds in arid environments (Godínez-Álvarez, Valiente-Banuet & Rojas-Martínez, 2002). The deposition of seeds under the cover of trees or shrubs (potential nurse plants) was estimated by focal observations to record the number of visits to these perching sites by birds after fruit consumption. To facilitate the monitoring of birds after they finished eating and departed to fly in another direction or roost on another plant, one person was dedicated to post-consumer observations.

The number of visits of frugivorous birds to each of the following categories of perch plants was recorded: (1) conspecific, indicating that the individual remained in the same plant species (i.e., B. longipes) where they ate fruit; (2) Fabaceae species, including trees and shrubs of the Caesalpinoideae, Faboideae and Mimosoideae subfamilies, which have been identified as potential nurse plants in semi-arid environments (Godínez-Álvarez & Valiente-Banuet, 1998) or (3) other tree or shrub plants, including species of Opuntia or columnar Cactaceae. Focal observations ended when eye contact with observed birds was lost. A contingency table of χ2 was used to compare the number of bird visits to each category of perch plant. The null hypothesis would be indicated by an equal number of bird visits among all perch categories across the three successional stages. Standardized residuals were used to evaluate the preferential use of certain perching sites by birds (Valiente-Banuet et al., 1991; Godínez-Álvarez, Valiente-Banuet & Rojas-Martínez, 2002). These residuals are distributed around a mean of 0 with a standard deviation of 1, and therefore, any resulting value ≥ 2 (approximately 5% of the normal distribution) was considered to be a significant deviation.

The probability that seeds were deposited in potentially suitable sites (under Fabaceae) was determined by the proportion of frugivore visits to these perch plants in relationship to the total number of recorded visits. Fabaceae plants have been shown to provide appropriate conditions for the recruitment of seedlings (Godínez-Álvarez & Valiente-Banuet, 1998).

Contribution of birds to seedling establishment in different successional stages

Two plots with a radius of 30 m (2,828 m2 per plot) were randomly chosen in each successional stage. In each plot seedlings and young individuals of B. longipes (height < 50 cm) were counted under trees or shrubs used by birds to roost after ingesting B. longipes fruits. Recruited individuals were classified by the previously mentioned categories of nurse plants; these categories have been successfully used in others studies in semiarid forests (Valiente-Banuet et al., 1991; Godínez-Álvarez, Valiente-Banuet & Rojas-Martínez, 2002). The number of young B. longipes plants observed under nurse plants was compared to the number of individuals expected to be recruited at random, derived from examining a proportional and comparable area and counting B. longipes underneath all plants with a DBH ≥ 10 cm (Valiente-Banuet et al., 1991). The null hypothesis would indicate a proportional number of seedlings between the comparative plots, given the coverage of woody plants. Standardized residuals were calculated to analyze the significance. Plant cover was determined in a previous study corresponding to the study sites (Almazán-Núñez et al., 2012).

Finally, each bird species was assigned a value of 0–1 according to their contribution towards the establishment of B. longipes. This value was estimated from individual observations of bird species after feeding on B. longipes drupes, their flight destination and number of visits to other plants. The maximum value was assigned to birds with the highest number of flights to nurse plants under which the largest number of seedlings or young B. longipes individuals had been observed with respect what would be expected by chance, according to the standardized residuals for each plot.

Results

Quantitative component

Frequency of visits and number of removed fruits

A total of 20 bird species were recorded eating B. longipes fruits (Table 1). Frequency of visits to remove fruit varied between stages (X2 = 54.78, df = 38, p < 0.05). Myiarchus tyrannulus and Tyrannus verticalis were the most frequent visitors to the early successional stage (Table 1), T. vociferans and T. verticalis to the intermediate stage and M. cinerascens to the mature stage. Spinus psaltria removed the greatest number of fruit at the early (5.00 ± 1.58) and intermediate (4.40 ± 0.51; Table 1) stages and Eupsittula canicularis at the mature stage (11.00 ± 4.00).

Table 1 Frequency of visits (visits/hr-observation), fruits and time spent per visit of birds that consumed B. longipes fruits in three successional stages of TDF in the Balsas basin of Guerrero, Mexico.

Family	Species	Early succession	Intermediate succession	Mature succession	
Freq. visit	Fruits/visit	Time/visit	Freq. visit	Fruits/visit	Time/visit	Freq. visit	Fruits/visit	Time/visit	
Psittacidae	Eupsittula canicularis	–	–	–	–	–	–	0.029	11.00 ± 4.00	8.00 ± 4.00	
Picidae	Melanerpes chrysogenys	–	–	–	0.057	2.25 ± 0.25	2.75 ± 0.48	0.071	2.66 ± 0.56	2.83 ± 0.53	
Tyrannidae	Myiarchus tuberculifer	–	–	–	0.043	2.00 ± 0.58	2.40 ± 0.83	0.086	1.83 ± 0.31	4.17 ± 0.65	
	M. cinerascens	0.086	2.50 ± 0.43	3.33 ± 0.99	0.257	2.11 ± 0.42	2.56 ± 0.37	0.243	2.52 ± 0.37	4.07 ± 0.34	
	M. nuttingi	–	–	–	0.143	2.20 ± 0.53	4.00 ± 0.68	0.1	2.77 ± 0.62	4.17 ± 0.76	
	M. tyrannulus	0.129	3.77 ± 0.52	5.00 ± 0.76	0.143	2.40 ± 0.37	4.90 ± 1.22	0.114	1.75 ± 0.49	2.00 ± 0.46	
	Myiodynastes luteiventris	0.029	3.50 ± 1.50	2.75 ± 0.25	–	–	–	0.071	4.33 ± 0.56	5.50 ± 0.99	
	Tyrannus melancholicus	–	–	–	0.014	5	10	–	–	–	
	T. vociferans	–	1.50 ± 0.50	1.75 ± 0.25	0.3	3.00 ± 0.43	6.33 ± 1.13	0.214	2.56 ± 0.52	4.41 ± 0.60	
	T. verticalis	0.129	4.11 ± 0.98	6.78 ± 1.30	0.3	3.52 ± 0.59	5.76 ± 1.02	0.129	5.45 ± 0.76	6.31 ± 1.13	
Corvidae	Calocitta formosa	–	–	–	0.029	3.00 ± 1.00	2.50 ± 0.50	–	–	–	
Vireonidae	Vireo gilvus	–	–	–	–	–	–	0.029	1.00 ± 0.00	1.50 ± 0.50	
Cardinalidae	Passerina caerulea	0.043	1.00 ± 0.00	1.83 ± 1.09	0.029	1.50 ± 0.50	5.00 ± 2.00	–	–	–	
	P. versicolor	0.029	1.00 ± 0.00	3.00 ± 2.00	0.114	2.38 ± 0.46	4.31 ± 1.02	–	–	–	
	P. leclancherii	–	–	–	0.029	2.00 ± 0.00	3.50 ± 0.50	0.029	1.50 ± 0.50	5.00 ± 1.00	
	Pheucticus melanocephalus	–	–	–	–	–	–	0.043	3.33 ± 1.86	6.83 ± 4.28	
Emberizidae	Haemorhous mexicanus	–	–	–	0.043	3.00 ± 0.00	3.67 ± 0.88	–	–	–	
Icteridae	Icterus wagleri	0.029	2.00 ± 0.00	2.00 ± 0.00	0.029	2.50 ± 1.50	5.00 ± 2.00	0.029	5.00 ± 1.00	3.25 ± 1.75	
	I. pustulatus	0.086	3.83 ± 0.87	3.33 ± 0.80	0.257	2.17 ± 0.26	3.27 ± 0.49	0.086	4.83 ± 1.33	8.50 ± 1.72	
Fringillidae	Spinus psaltria	0.057	5.00 ± 1.58	4.25 ± 1.16	0.071	4.40 ± 0.51	5.00 ± 0.89	0.029	2.00 ± 0.58	2.83 ± 0.17	
Notes:

The values shown are as the mean ± standard error.

The species with a dash (–) were not observed visiting trees.

Overall, 17.9% of the fruits consumed in the three stages were removed at the early stage, 42.2% at the intermediate stage and 39.9% at the mature stage (n = 825), although no significant differences were found among stages (F2,275 = 1.57, p = 0.210). The flycatcher T. verticalis remained for the longest time in the trees of the early stage (6.78 ± 1.30 min), T. vociferans in the intermediate stage (6.33 ± 1.13 min) and E. canicularis in the mature stage (8.00 ± 4.00 min; Table 1).

Seed germination

The shortest average seed retention time from fruit intake until evacuation was recorded for Myiarchus nuttingi and the highest for M. tyrannulus (Table 2). The latter had the widest range in seed evacuation time (minimum = 10 min and maximum = 230 min). The shortest average timeframe corresponded to Myiodynastes luteiventris (minimum = 12 min and maximum = 155 min; Table 2), although differences in retention time were not significant (F4,122 = 0.98, p = 0.420). Body size of the frugivorous birds was positively correlated with time of passage of B. longipes seeds (R2 = 0.79, F = 11.68, p = 0.04).

Table 2 Average seed retention time from point of seed consumption to defecation by captured birds.

Statistics	Myicin	Myinut	Myityr	Myilut	Melchr	
Average time (min)	104	60	129	69	80	
Standard Error	11.3	5.8	23.0	8.2	11.0	
Minimum	22	18	10	12	7	
Maximum	225	179	230	155	155	
Note:

Myicin (Myiarchus cinerascens), Myinut (Myiarchus nuttingi), Myityr (Myiarchus tyrannulus), Myilut (Myiodynastes luteiventris), Melchr (Melanerpes chrysogenys).

None of the seeds with intact pseudoaril (control group 1) germinated (Fig. 2). Seeds without pseudoaril (control group 2) had a germination rate of 10%. The seeds that passed through the gut of Myiarchus cinerascens had the highest germination percentage (27%, n = 33), followed by Myiarchus tyrannulus (26%, n = 27), Melanerpes chrysogenys (24%, n = 29), Myiarchus nuttingi (15%, n = 67) and Myiodynates luteiventris (12%, n = 58) (Fig. 2). According to GLM, seed germination varied among treatments (X2 = 21.73, p = 0.001). Post hoc pairwise t-test comparisons indicated that the three bird species associated with the highest percentage of germination (M. cinerascens, M. tyrannulus and M. chrysogenys) significantly differed in comparison to seeds without pseudoaril (control group 2, p < 0.01) (Fig. 3). However, significant differences were not found in germination between seeds without pseudoaril (control group 2) and seeds eaten by M. nuttingi and M. luteiventris. The passage time of seeds through the gut and the germination percentage were marginally significant (R2 = 0.71, F = 7.69, p = 0.06). Body size was not significantly correlated with germination percentage (p > 0.05).

Figure 2 Seed germination of B. longipes after passing through the digestive system of birds in comparison to control group 1 (seed with pseudoaril) and control group 2 (seeds without pseudoaril).

Figure 3 Proportion of seeds germinated after passing through the digestive system of birds.

Different letters among treatments indicate significant differences (T-test comparisons, p < 0.05).

Potential suitable sites for recruitment

After consuming fruits, birds perched in three categories of plants (Fig. 4). The preference was for Fabaceae species at all three stages (X2 = 22.98, df = 12, p < 0.05; X2 = 55.33, df = 20, p < 0.05; X2 = 54.98, df = 20, p < 0.05, for the early, intermediate and mature stages, respectively) (Figs. 4A–4C). At the intermediate and mature stages, flycatchers M. nuttingi and M. tuberculifer remained for the longest period of time in Acacia and Mimosa plants following feeding episodes, while Tyrannus verticalis and Vireo gilvus spent more time in the same trees where they feeded on fruits. Thus, flycatchers of the Myiarchus genus were the most likely species to deposit B. longipes seeds beneath Mimosa and Acacia trees and shrubs across the three successional stages (Table 4).

Figure 4 Residuals of a contingency table comparing perching sites for birds after consumption of B. longipes fruit in three stages of succession: (A) early, (B) intermediate and (C) mature.

Bars represent frequencies, and significant residuals indicate preference (positive residual) or avoidance (negative residual) by each bird species. * p < 0.05, ** p < 0.01.

Contribution of birds to sites of seedling establishment

The lowest density of B. longipes seedlings and non-reproductive individuals was found at the early stage (0.002 ind/m2). The average height of plants was 54.07 ± 7.90 cm. At the intermediate and mature stages, densities of seedlings and non-reproductive individuals were 0.007 ind/m2 and 0.008 ind/m2, with an average height of 50.93 ± 3.90 cm and 53.19 ± 3.80 cm, respectively, and did not differ significantly in density (F2,5 = 0.89, p = 0.50) or in average height (F2,104 = 0.12, p = 0.89).

At the early stage, the number of seedlings and young B. longipes individuals was significantly higher under Tecoma stans, Plocosperma buxifolium and Mimosa polyantha plants (Table 3). At the intermediate stage, the largest number of seedlings was found under Mimosa polyantha and Calliandra eryophylla, and at the mature stage, under Eysenhardtia polystachya, Senna wislizeni, Sebastiana pavoniana and Acacia cochliacantha (Table 3). Acacia subangulata was the only legume that presented a lower number of observed seedlings than expected by chance (Table 3).

Table 3 Standardized Residuals (StaRes) for the number of B. longipes seedlings (< 50 cm) according to observed (Obs) and expected (Exp) coverage under nurse plants including two categories: other tree and shrub species and Fabaceae species.

Plant species	Family	Early stage	Intermediate stage	Mature stage	
Obs	Exp	StaRes	Obs	Exp	StaRes	Obs	Exp	StaRes	
Other tree and shrub species	
Tecoma stans	Bignoniaceae	1	0.1	3	1	0.3	1.3	0	0	−0.2	
Cordia sp	Boraginaceae	1	1.9	−0.7	0	1.2	−1.1	1	0.8	0.3	
Opuntia sp	Cactaceae	0	0.2	−0.4	1	0.2	1.6	0	0	0	
Ipomoea pauciflora	Convolvulaceae	0	0.5	−0.7	0	2.7	−1.6	1	1.2	−0.1	
Euphorbia schlechtendalii	Euphorbiaceae	0	0.1	−0.3	0	2.4	−1.6	1	0.7	0.4	
Sebastiana pavoniana	Euphorbiaceae	0	0	0	0	0	0	1	0.2	2.2	
Plocosperma buxifolium	Loganiaceae	1	0.1	3.1	0	2.7	−1.6	1	0.6	0.5	
Ziziphus amole	Rhamnaceae	0	0	0	0	0	0	1	0.8	0.2	
Randia sp	Rubiaceae	0	0	0	1	0.3	1.2	0	0	0	
Cissus sp	Vitaceae	1	1.3	−0.2	6	3	1.8	0	0.5	−0.7	
Fabaceae species	
Senna wislizeni	Fabaceae	2	0.7	1.5	3	1.1	1.8	8	2.9	3	
Senna skinneri	Fabaceae	0	0	0	2	1.2	0.8	0	0.3	−0.5	
Eysenhardtia polystachya	Fabaceae	0	0.5	−0.7	0	3.5	−1.9	2	0.2	4.4	
Gliricidia sepium	Fabaceae	3	3.5	−0.3	0	3.3	−1.8	0	0.8	−0.9	
Havardia acatlensis	Fabaceae	2	1.5	0.4	3	4.3	−0.6	1	2.7	−1	
Pterocarpus acapulcensis	Fabaceae	0	0.3	−0.6	2	4.1	−1	3	8.5	−1.9	
Acacia cochliacantha	Fabaceae	1	1.4	−0.3	2	0.7	1.5	9	3.1	3.3	
Acacia subangulata	Fabaceae	0	0.9	−1	3	6.1	−1.3	5	13.6	-2.3	
Calliandra eryophylla	Fabaceae	0	0	0	4	0.7	4	0	0	0	
Desmanthus balsensis	Fabaceae	0	0	0	0	0.7	−0.9	3	3.5	−0.2	
Lysiloma tergemina	Fabaceae	0	0.5	−0.7	5	2.5	1.5	2	2.2	−0.2	
Mimosa polyantha	Fabaceae	2	0.4	2.3	10	1.9	5.9	9	4.6	2.1	
Notes:

Residual values > 2 are significant at a 95% confidence level for a normal distribution.

Significant values of standardized residuals (StaRes) for each successional stage are highlighted in bold.

The largest contribution to the establishment of B. longipes seedlings, calculated based on the number of flights to nurse plants with the largest number of observed seedlings with respect what would be expected by chance, was attributed to M. cinerascens at the early stage and to M. nuttingi at the intermediate and mature stages (Table 4).

Table 4 Probability of seed deposition, contribution to the establishment of seedlings and effectiveness of B. longipes seed dispersal by frugivorous birds in a successional gradient of TDF in the Balsas basin of Guerrero, Mexico.

Species	Probability of B. longipes seed deposition in suitable sites	Value of contribution to the establishment of B. longipes seedlings in suitable sites	Seed dispersal effectiveness	
Early	Intermediate	Mature	Early	Intermediate	Mature	Early	Intermediate	Mature	
C. formosa	–	0.02	–	–	–	–	–	–	–	
I. pustulatus	0.15	0.07	0.05	0.16	0.6	–	–	–	–	
I. wagleri	0.05	0.01	0.02	–	0.2	–	–	–	–	
M. chrysogenys	–	0.02	0.06	–	0.2	0.25	0	0.015	0.08	
M. cinerascens	0.29	0.18	0.17	1	0.2	0.5	1.69	0.52	1.72	
M. luteiventris	0.05	–	0.09	–	–	0.25	0	0	0.10	
M. nuttingi	–	0.18	0.09	–	1	1	0	0.84	0.48	
M. tuberculifer	–	0.12	0.08	–	–	0.25	–	–	–	
M. tyrannulus	0.20	0.14	0.08	–	0.4	0.25	0	0.51	0.10	
T. melancholicus	–	0.02	–	–	–	–	–	–	–	
T. verticalis	0.17	0.14	0.11	–	0.4	0.25	–	–	–	
T. vociferans	0.10	0.08	0.25	–	–	0.50	–	–	–	
V. gilvus	–	–	0.01	–	–	–	–	–	–	

Seed dispersal effectiveness

The effectiveness of seed dispersal was estimated for five bird species whose number of visits allowed for a reliable estimation, which varied depending on the stage (Table 4). For other species, dispersion was not determined due to lack of defecated seeds or other subcomponents that would allow for this assessment.

At all stages the best dispersers belonged to the genus Myiarchus. At the early stage, only M. cinerascens contributed to seed dispersion (Table 4). At the intermediate stage, M. nuttingi was the largest contributor to seed dispersion. In the mature stage, five species participated in seed dispersion, and M. cinerascens had the highest effectiveness (Table 4).

Discussion

Bursera longipes fruits were consumed by birds in all successional stages, although bird species participating in seed dispersal, their importance and plants used for perching after feeding varied among stages. Both the number of fruit-eating species and the frequency of bird visits were lower in the early successional stage. This result concurs with the reports of Cordeiro & Howe (2003) and Figueroa-Esquivel et al. (2009), whom also note that at disturbed sites, the number frugivorous bird species and their frequency tends to decrease due to a lower supply of resources. Since lower number of frugivores are found in early-successional stages, there is less redundancy in their dispersal community. Also, bird assemblages between seral stages may be complementary because there are differences in functional diversity, which is mainly driven by their differences in movement, foraging behavior, as well as body size (Calviño-Cancela & Martín-Herrero, 2009; Escribano-Avila et al., 2014).

Although several bird species removed many fruits and consistently visited B. longipes trees at all three stages, not all birds contributed to the effective dispersal of its seeds. For example, Spinus psaltria and Eupsittula canicularis had the highest rate of fruit removal at all three stages. S. psaltria chews the seeds’ pseudoaril and later discards seeds under the parent plant, which reduces chances of germination and establishment due to competition with other conspecifics (Jordano & Schupp, 2000; Bas, Pons & Gómez, 2005). In fact, we did not observe juvenile plants of B. longipes growing in association with parent plants. E. canicularis destroyed or damaged seeds upon handling them, and thus their contribution to dispersion was negative. These results are consistent with those observed in other Neotropical plants, where the rate of seed mortality increases due to predation by Psittacidae species (Francisco et al., 2008).

Previously, it was suggested that species of the Tyrannidae family, particularly from the Myiarchus genus, could be the main legitimate dispersers of Bursera fruits, despite being largely insectivorous (Ramos-Ordoñez & Arizmendi, 2011). In our study, ca. 70% of the seeds at the three stages were removed by Tyrannidae. Myiarchus spp. individuals, whose behavior is less gregarious compared with other birds that also consume B. longipes fruits (e.g., genera Tyrannus, Eupsittula, Spinus) and who removed about 30% of fruits at all stages. Two of these species are latitudinal migrants (M. cinerascens and M. nuttingi), and other two have local altitudinal migrations (M. tuberculifer and M. tyrannulus).

In the case of the M. cinerascens and M. nuttingi, the ripening time of B. longipes fruits coincides with the winter presence of these birds. In winter, energy demands increase due to the long distance movements of these species (Tellería, Ramírez & Pérez-Tris, 2005), and the fruits of Bursera spp. represent a source of lipids, which are present in the pseudoaril overlaying the seeds (Ramos-Ordoñez, Arizmendi & Márquez-Guzmán, 2012).

The distribution pattern of M. tuberculifer and M. tyrannulus apparently is determined by supply of Bursera fruits (Almazán-Núñez et al., 2015), as the two bird species were only present at study sites during the fruiting season. These flycatchers were also observed during fruiting of B. morelensis in the Tehuacan Valley in Puebla (Ramos-Ordoñez & Arizmendi, 2011). This is similar to the white-eyed vireo (Vireo griseus), whose presence and abundance was previously correlated with number of B. simaruba fruits in secondary growth forests in the Yucatan Peninsula (Greenberg, Foster & Marquez-Valdelamar, 1995).

The distribution of Bursera spp. and particularly B. longipes seemingly coincides with that of the Myiarchus genus throughout the Neotropics, and in this study, these flycatchers have proven to be its most effective seed dispersers (sensu Schupp, 1995). The distribution of both groups is characteristic of semi-arid environments in the Neotropics, and these birds and the plant genera diversified during the Miocene (Ohlson, Fjeldså & Ericson, 2008; De-Nova et al., 2012). Both groups also generally adapt to anthropogenic disturbances throughout their range (Howell & Webb, 1995; Rzedowski, Medina Lemos & Calderón de Rzedowski, 2005), and this may also be a determinant factor of their recent success throughout Mesoamerican tropical forests.

Overall, the minimum retention time of seeds by frugivorous seed dispersers was greater than the time they remain on B. longipes trees. This indicates that birds do not defecate immediately after feeding, and therefore seeds are transported and deposited to other sites relatively far away from the mother plant, such as under nurse plants (Schupp, 1995; Godínez-Álvarez & Valiente-Banuet, 1998; Padilla & Pugnaire, 2006).

Moreover, the germination rate of seeds that passed through the gut of M. cinerascens, M. tyrannulus and M. chrysogenys was significantly higher than seeds without pseudoaril (control group 2), although not in the case of M. nuttingi and M. luteiventris. While Bursera seeds responded to similar physiological treatments during endozoochory (Stevenson et al., 2002), germination differences between species can potentially be explained by time spent in the digestive system (Domínguez-Domínguez, Morales-Mávil & Alba-Landa, 2006). M. nuttingi and M. luteiventris had a lower retention time; although these differences were not significant, there was a marginally positive significant relationship between gut passage time and germination, as found in other studies (Traveset, 1998). It is necessary to clarify that these results should be taken with caution because retention times represent an approximation. On the other hand, the size of the birds does not explain the percentage of germination. It is therefore possible that certain aspects of the digestive physiology of each bird species are more important than bird size in the subsequent germination of seeds. For example, some passerine birds retain the seeds in the gut for much longer than other birds (M. cinerascens, mean 129 min, versus M. chrysogenys, mean 80 min), as also found by Jordano (1992). These bird species consume insects throughout much of the year, but since their intestines are usually small, additional enzymatic attack is required; in contrast to strict frugivores whose digestive system is usually longer (Jordano, 1986).

Bursera longipes seeds are hard and possibly require a longer digestion time in order to soften the endocarp. As none of the seeds with intact pseudoaril germinated, the importance of frugivorous birds in removing this tissue may be highlighted (Robertson et al., 2006), as these tissues may contain substances that inhibit seed germination, similar to B. fagaroides (Ortiz-Pulido & Rico-Gray, 2006). In this sense, Robertson et al. (2006) also indicated that the deinhibition process may be more important than scarification of seeds. However, both phases (deinhibition and scarification) are generally important for B. longipes seeds, and in general for Bursera species, as shown in other studies (Ortiz-Pulido & Rico-Gray, 2006; Ramos-Ordoñez & Arizmendi, 2011).

The probability that B. longipes seeds are deposited by flycatchers of the genus Myiarchus in suitable sites for germination, as well as the subsequent development of seedlings, confirms that these flycatchers are the most efficient seed dispersers across different successional stages of dry forest. These dispersers often select perches belonging to species of Fabaceae, which have been previously identified as nurse plants throughout several arid environments of Mexico (Valiente-Banuet et al., 1991; Godínez-Álvarez & Valiente-Banuet, 1998; Castillo Landero & Valiente-Banuet, 2010). After eating fruits of B. longipes, Myiarchus species remain in the tree for a brief duration of time and then generally move short distances to potential Fabaceae nurse plants near Bursera trees. This behavior has been observed in other frugivorous birds (Westcott & Graham, 2000). In fact, in this study several species of Fabaceae (Senna wislizeni, Eysenhardtia polystachya, Acacia cochliacantha, Calliandra eryophylla and Mimosa polyantha) were important for the recruitment of B. longipes seedlings. This demonstrates the importance of legumes in aiding seedling establishment, as they foster necessary conditions due to their recycling of nutrients, accumulation of organic matter and shadow effect, which leads to more favorable temperatures for native plant development (Franco & Nobel, 1989; Bonanomi et al., 2007). In addition, the spines of many of these species help to dispel potential predators of seedlings (Khurana & Singh, 2001).

On the other hand, only M. polyantha had more B. longipes seedlings in its vicinity than expected by chance across the three successional stages, which suggests that in addition to presence of nurse plants, other factors (e.g., soil water content, bulk density of soil, and pH) could be limiting the recruitment of B. longipes along the successional gradient (Buzzard et al., 2015). Therefore, the quality of microhabitats depends not only on presence of Fabaceae species but also on biotic and abiotic factors that can limit seedling recruitment (Castillo Landero & Valiente-Banuet, 2010).

Other members of the Tyrannidae family, such as T. verticalis and T. vociferans, have been also considered to be dispersers, since they remove fruits in large quantities and swallow whole seeds (Almazán-Núñez et al., 2015). However, they typically perch on the top branches of B. longipes trees for long periods of time, and therefore, the quality of dispersal by these species is low, because B. longipes crown does not seem to be suitable for recruitment of conspecific seeds.

The observed number of recruited seedlings was lower in the early successional stage in comparison to intermediate and mature stages. Mimosa polyantha was one of the nurse species preferred by dispersers for perching; resulting in a higher recruitment of seeds, and consequently, the number of seedlings under this plant was higher than expected by chance at all successional stages. Similar results have been obtained for other species of the same nurse plant genus (Castillo Landero & Valiente-Banuet, 2010).

At the mature and intermediate stages, density of recruited seedlings increased due to seed rain from dispersers, as found in other Neotropical forests (Vasconcellos-Neto, Barbosa de Albuquerque & Rodrigues Silva, 2009). The seed bank of the soil is also a likely influence and may have a lower density in earlier successional stages, as found at other TDF sites (Uhl, 1987; Hammond, 1995). The presence of a seed bank fosters a higher rate of germination at advanced successional stages, which in addition to a high number of disperser visits would improve the microenvironmental conditions favoring seedling establishment (Valiente-Banuet et al., 1991; Godínez-Álvarez & Valiente-Banuet, 1998; Orozco-Almanza et al., 2003; Padilla & Pugnaire, 2006).

Although the number of frugivorous birds was lower in earlier successional stages (Almazán-Núñez et al., 2015), the presence of migratory dispersers that can tolerate disturbed sites ultimately contributes to the regeneration of these forests (Griscom, Kalko & Ashton, 2007; Tellería, Carrascal & Santos, 2014). Despite lower densities, seedlings recruited under nurse plants in the early stage were larger in size than seedlings found at other stages. However, it is also likely that the process of germination and establishment at this stage is slower than at other successional stages, mainly due to the more inhospitable conditions faced by seeds once they are dispersed (Padilla & Pugnaire, 2006; Obeso, Martínez & García, 2011). In this scenario, greater presence of legumes at early successional stages, as well as the preference of various dispersers to perch on these plants and the adaptability of some zoochorous plants to these new conditions, leads to a greater chance of recovering these habitats. Regional or genetic studies are necessary in order to analyze the spatial variability of seed dispersal and to further understand both the preferences and movements of frugivorous birds (Carlo, Aukema & Morales, 2007; González-Varo, Arroyo & Jordano, 2014), as well as differences in these factors in distinct regional environments.

The TDF ecosystem in Mexico and Central America is expected to reduce by more than 60% in the next 40 y, according to scenarios of climate change (Miles et al., 2006; Prieto-Torres et al., 2015). As a consequence, changes may occur in the fruiting phenology of trees that could uncouple interactions with dispersers. Furthermore, climate changes could modify the movement patterns of migratory birds, which are efficient dispersers of B. longipes seeds and other species (Santos & Tellería, 1995; Tellería, Carrascal & Santos, 2014).

We thank Hector Godínez, Pedro Luis Valverde and Maria Ramos Ordoñez for the critical review of this document, as well as two anonymous reviewers who improved this manuscript. We also appreciate the valuable support during fieldwork by Roberto Bahena, Trinidad Cruz, Noemí González, Jeraldín González, Brenda Pérez, Pablo Sierra and Jaili Sánchez.

Additional Information and Declarations

Competing Interests

Author Contributions

Animal Ethics

Data Deposition

Luis E. Eguiarte is an Academic Editor for PeerJ.

R. Carlos Almazán-Núñez conceived and designed the experiments, performed the experiments, analyzed the data, contributed reagents/materials/analysis tools, wrote the paper, reviewed drafts of the paper, financial and logistic support, prepared figures and/or tables.

Luis E. Eguiarte conceived and designed the experiments, contributed reagents/materials/analysis tools, reviewed drafts of the paper, financial support.

María del Coro Arizmendi conceived and designed the experiments, contributed reagents/materials/analysis tools, reviewed drafts of the paper, financial and logistic support.

Pablo Corcuera conceived and designed the experiments, performed the experiments, analyzed the data, contributed reagents/materials/analysis tools, wrote the paper, prepared figures and/or tables.

The following information was supplied relating to ethical approvals (i.e., approving body and any reference numbers):

The techniques used to collect vomit and feces were non-invasive and it was not necessary to have a special authorization.

The following information was supplied regarding data availability:

Database birds (Bursera): https://figshare.com/s/56f8a587a3813cea257c.

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
