# Peer review of "Myiarchus flycatchers are the primary seed dispersers of Bursera longipes in a Mexican dry forest"

_PeerJ, doi:10.7717/peerj.2126_

## Round 0.1 · original submission · Major Revisions

You will see that while the reviewers find your work of interest, reviewer #1 raised some concerns regarding experimental design and the approach used to estimate some of the variables of the study. These raised points need to be addressed before a decision on publication can be made.

·

Basic reporting

I think the manuscript by Almazán-Núñez et al. overall fulfill the basic reporting criteria of PeerJ. Only some point will need further revision. This is: "The article should include sufficient introduction and background to demonstrate how the work fits into the broader field of knowledge" I think some additional effort should be done to contextualize and introduce the questions made by the authors. Additional comments and suggestions are provided in the "General comments section".

Experimental design

This paper describes primary research within the aims and scope of PeerJ. The paper defines research questions and the gap of knowledge that will contribute to fill. However there are some concerns regarding experimental design, estimation of some relevant variables and also some basic information that allow the repeatability of the study is lacking. No ethical issues have been detected. Additional information on how to deal with these issues is provided in "General comments section".

Validity of the findings

Some concerns regarding the estimation of some relevant variables have been detected and also some over interpretation in the conclusion section. Data are not referred to be in an acceptable repository. Authors should make their data available in order to publish their work according to the journal policy. Despite this work fails to satisfy some of the points asked by the journal I think the majority of these issues can be dealt with. In this regard I hope the suggestions and comments provided result helpful.

Additional comments

General comments

This is an interesting study that provide information on the seed dispersal (and partial seed dispersal effectiveness components) generated by the avian disperser community of B. longipes a dominant species of neotropical dry forests a greatly threaten ecosystem. This work has special merit as very little empirical information on the dispersal & recruitment patterns of endozoochorus plant species is available for tropical dry forests. Authors evaluate how such patterns vary in a disturbance gradient. However here I detect the first problem with the experimental design. Three sites are studied and each of the sites represents a differential succesional stage “early, intermediate, mature”. Thus the factors “site” and “succesional stage” are confounded. This should be acknowledged and discussed accordingly along the manuscript.

My second main concern is related to how authors determine “safe sites”, the use birds may do of safe sites and the way in which authors estimate such variables. Thus in the formula used to index seed dispersal effectiveness the terms “seed deposition probability” and “contribution of birds to the site of establishment” are estimated in such a way that the estimates obtained do not result reliable and robust for the purpose of determining SDE for the avian community. I explain myself below.

Seed deposition probability
Seed deposition probability is estimated by means of “the number of visits to potential nurse plants” (Lines# 231 a 240). Authors are assuming a direct and positive relationship between birds perch frequency and the quantity of seeds being delivered underneath such perches, despite being possible that such a relation exists it is not proved not even supported with literature. In this case, variables related to the time spent in each kind of perch would be a better estimate, (but still quite coarse). Additionally authors’ are assuming equal availability and detectability of birds perching in the different kind of perches around the focal points; however none of these is proved. Thus I think results may be quite bias by the abundance and spatial location of perches around the twenty one (7x3) Bursera trees in which the observations have been performed.

Link between seed deposition probability microhabitats and microhabitat quality
If I understood well the approach authors are following they try to determine deposition probability as explained above and then relate such deposition patterns with the “quality of each microhabitat” however the categories used in each of the tasks do not match. To determine deposition patterns they used the categories (Acacia, Mimosa, other tree, other shrub and Bursera, Fig. 4). In the case of “the quality of microhabitat” they estimate the abundance of Bursera seedlings in relation to the cover of each microhabitat, that in this case were different species (Table 3) that may belong to more than one of the categories established to determine deposition patterns.
Additionally, in my opinion, it is not clear why the categories established (Acacia, Mimosa, other tree, other shrub and Bursera) are relevant from the SDE approach. For example, authors’ results show that microhabitat quality is highly site-dependent (Table 3), and it is not clear what it is behind such “quality”. For instance, if we think on soil nutrients we would expect legume species to provide with higher quality, however legume species were not coherently a “better” microhabitat (see Table 3) despite what authors discuss later (see Lines# 427-430).
Microhabitat quality should account for both biotic (e.g. post-dispersal seed predation, pathogen abundance…) and abiotic (soil moisture, water..) factors that may limit plant recruitment. From my point of view, given the observational approach performed, information about how those microhabitats contribute or hamper seeds recruitment (e.g. by protecting from seed predators, low abundance of pathogen, adequate abiotic environmental conditions) is needed in addition to the referred literature in which acacia species acted as nurse species in other study area and for other plant species.
Overall, regarding this point I think the data provided here are insufficient to determine quantitative deposition patterns of birds in different microhabitats and thus I do not think this study can provide SDE estimates. Instead I suggest authors to download their results and use the data collected adequately.
I propose authors’ to establish clearly matching categories between perches visited by birds and microhabitat quality and given that such data is not directly linked I would used such information in the discussion to speculate about the possible relationships that may be between birds movement patterns and the microhabitat in which more recruitment is observed.

From my point of view, the strong point of this work is the quantitative component and the (partial) qualitative component of the gut passage effect (however see the comment below). I Think authors may provide further and interesting insides focusing in this two areas. For instance I would found interesting to know how functional traits of birds vary along the studied sites and if may be some functional traits such as body size or beak width, relate to the quantity of seeds dispersed and the quality birds provide to dispersed seeds. For instance I think the relationship between gut passage time and germination was not evaluated and I think that would be interesting. A positive relation between body size, gut passage time and germination improvement could be predicted (Traveset 1998) and tested in this case.
Another point I think it may be interesting to discuss is that given that a remarkable number of bird species involved in promote dispersal patterns are migrant and that tropical dry forest are highly threaten by climate change (Miles et al. 2006). The fruiting phenology and movement patterns of B. longipes and several members of its dispersal community may result uncoupled and have different consequences for both dispersal patterns of the plants and the avian community (Santos & Tellería 1995, Tellería et al. 2014).

Unfortunately I have another concern related with the gut passage effect experiment. In Line# 214-215 it is said that 50 seeds per site were used for each of the controls, fruits with aril and fruits with aril). If what authors did was to collect 50 fruits in site “1” and 50 fruits in “site 2” and used them separately for controls 1 and 2 I’m afraid they have again a confounding effect of site and control. Ideally they should have used half of each seeds for each treatment or full mixed then and assigned then randomly to each control treatment. Given that it could be expectable that fruits collected in the “mature site” have better germination rates than those collected at the “intermediate succession site” if this is the case I’m afraid no inference can be done regarding the two control treatments. In this case it may be better to remove the control treatment 1 (with the aril) and only compare results with the control treatment without the aril. This however may be a pity as very few germination experiments dealing with gut passage effect adequately evaluate the deinhibition and the scarification effect (Robertson et al. 2006) something that could have been done in this work with the adequate experimental design.

I provide some minor comments below, not much in detail, as I think that additional work of re-writing and re-interpreting should be done before this manuscript can be re-considered for publication.

INTRODUCTION
L# 53, what do you mean by new sites? Sites where the species being disperse is not present? Please be more specific.
L# 60-62. Dispersers may do this, but they also may disperse seeds to unfavorable microhabitats (e.g. Calviño-Cancela & Martín-Herrero, 2009). Authors’ are referring to a particular situation that may or may not occur, I suggest re-writing accordingly.
L# 92-94. I don’t understand this sentence… I think something is missing.
L# 118-120. Questions 2, 3, 4 are related to the Seed Dispersal Quality Component (sensu Schupp et al. 2010). Dispersal quality of dispersers in the mouth and gut, dispersal pattern that determine the microhabitat of arrival of seeds…However this theme is not sufficiently introduced. Why are these questions relevant for B. longipdes? Why do you expect different dispersers to present differential effects due to gut passage? Do you expect a different deposition pattern that may affect seedling recruitment, why? I would expect these kinds of questions to be introduced in the introduction and I suggest re-writing it accordingly. Much attention is paid to the disturbance gradient and its importance due to the lack of studies dealing with wider spatial scales, something that I agree with, however the experimental design of this study, (this will be discussed later) unfortunately do not allow to say much about this.
I suggest including the information referred above, and thus, to avoid a too long introduction I suggest moving Lines# 98–107 to the methods section as the study species B. longipdes is already described in the introduction.

MATERIAL AND METHODS
L# 124 Additional information regarding the study sites is necessary. What extension was covered by the study sties? What was the distance between study areas? How did data were collected, by means of plots, transects? An additional sub-figure within figure 1 representing the study areas and its coordinates would be nice.
L# 199 How did you establish the time at which consumption started? And how do you relate consumed and defecated or regurgitated seeds? I think this section should be better explained.
L# 222 Given that the gut passage effect experiment was performed with a different amount of seeds for each bird species and control treatments a General Linear Model with proportional error distributions should be more adequate (Crawly, the R Book) to control for both the response variable itsef and also for the unbalanced nature of these data.
L# 289 Did you count the fruits? This is not explained before. If so, there were differences in fruit abundance among sites? This is a relevant variable to explain fruit removal patterns.

DISCUSSION
L# 361-364. I think this is one of the most interesting results of this manuscript. However the atomization of “succesional stage” and “site” limits the conclusions that can be inferred from these results. Something that can be pointed out here is if due to the reduction in the quantity of species that disperse seeds in the more disturbed site these areas may have experienced a reduced functional complementarity or redundancy. If this is the case the “mature” site present a functionally redundant community being the dispersal services robust to the loss of some bird species, instead in if the early succesional stages present a lower diversity of dispersers the dispersal services they provide may be more vulnerable to any additional disturbance (Escribano-Avila et al. 2014). Some discussion in this regard may be included however with the adequate acknowledgment of the limitation related with the no replication.
L# 369: Is this the case for this species? Do you know if B. longipes present negative density dependence? Did you find recruitment under parent trees?
L# 371-374. I do not understand this sentence. Please rephrase.
L# 384-385. The higher demand of lipids in the winter referred here by the authors is due to the lower temperatures experienced by several temperate species in the north hemisphere that additionally match with their reproductive period. I don’t know if this makes sense for the dry and tropical latitudes in which this study is performed.
L# 407 In seed dispersal studies gut passage is more common than digestive track. Change it accordingly along the manuscript.
L# 407-420. This part is one of the strongest points of this work, in my opinion. It would be nice to have a statiscal analyses relating gut passage time with germination rate. And additional insides discussing the differences between the deinhibiton and scarification effect (sensu Robertson et al. 2006) would be really interesting in case the experimental design allow it (See general comments).
L# 417-420 These lines seem to be out of context to me… Please remove or rephrase.
L# 443-450 Do B. longipdes forms a seed bank? This is not a very common strategy in endozoochorus plant species.
L# 454: But you previously said there is not recruitment underneath mother plants in this species (Lines# 434-435)??
Along the manuscript I have detected the use of DFT and TDF to refer to tropical dry forests. Please use always the same contracted form.
L# 466-470 I think this sentences are not appropriate given that this study can’t say much about this given its experimental design limitations.
L# 471-472 Then the patterns of recruitment detected for B. longipes may be due this abiotic variability…
L# 476-478 I don’t think that what authors say in this sentence is at all justified, is far too speculative with no evidence about co-evolutionary process in this article.

References
Calviño-Cancela, M., & Martín-Herrero, J. (2009). Effectiveness of a varied assemblage of seed dispersers of a fleshy-fruited plant. Ecology, 90(12), 3503-3515.
Crawley, M. J. (2012). The R book. John Wiley & Sons.
Escribano‐Avila, G., Calviño‐Cancela, M., Pías, B., Virgós, E., Valladares, F., & Escudero, A. (2014). Diverse guilds provide complementary dispersal services in a woodland expansion process after land abandonment. Journal of Applied Ecology, 51(6), 1701-1711.
Miles, L., Newton, A. C., DeFries, R. S., Ravilious, C., May, I., Blyth, S., ... & Gordon, J. E. (2006). A global overview of the conservation status of tropical dry forests. Journal of Biogeography, 33(3), 491-505
Robertson, A. W., Trass, A., Ladley, J. J., & Kelly, D. (2006). Assessing the benefits of frugivory for seed germination: the importance of the deinhibition effect. Functional Ecology, 20(1), 58-66.
Santos, T., & Tellería, J. L. (1995). Global environmental change to and the future of Mediterranean forest avifauna. In Global change and Mediterranean-type ecosystems (pp. 457-470). Springer New York.
Schupp, E. W., Jordano, P., & Gómez, J. M. (2010). Seed dispersal effectiveness revisited: a conceptual review. New Phytologist, 188(2), 333-353.
Tellería, J. L., Carrascal, L. M., & Santos, T. (2014). Species abundance and migratory status affects large-scale fruit tracking in thrushes (Turdus spp.).Journal of Ornithology, 155(1), 157-164.
Traveset, A. (1998). Effect of seed passage through vertebrate frugivores' guts on germination: a review. Perspectives in Plant ecology, evolution

Reviewer 2 ·

Basic reporting

No Comments

Experimental design

No Comments

Validity of the findings

No Comments

Additional comments

L. 89-91 – the authors reduce the causes underlying the geographic mosaic of coevolution to a question of species distribution, but a multitude of other causes, from the genetic profile of interacting populations to abiotic issues, might also be involved.
L. 120 – replace “dispersal birds” to “seed-dispersing birds”
L. 157 - The abundance of birds should not enter in the calculation of the quantity componente of seed dispersal effectiveness because it is not directly related to seed removal. Suppose an abundant species that for some reason is not frequent at fruiting trees. The inclusion of its abundance in the calculation of its effectiveness would inflate its contribution to seed dispersal. Therefore, there is no reason to analyze the difference in bird abundance among sites (L. 172-175).
L. 222 – Is the “time that seeds remained in the digestive system of birds” really analyzed in this way?
L. 241 – replace “with” to “to”
L. 243-247 – To really analyzed “preferential use by birds of certain perching sites” it is essential to evaluate the availability of the diferente perching plants and contrast it to the use that birds make of these plants.
L. 356 – replace “envolve” to “envolved”

---

## Round 0.2 · Minor Revisions

The reviewer has indicated a good number of suggestions aimed at improving the writing of some of the text. Clarification and justification of some of terminology used is also needed. Since the authors have already resubmitted once I suggest to them to address all points with enough clarity. Also it is important that the rebuttal letter provides enough explanation and details of the rationale for the answers to the reviewer.

·

Basic reporting

L 60 add "may" between dispersers and aid (because dispersers may also deposite seeds in unsuitable microhabitats).


L 60-63: I think these lines can be better explained by trying to make the idea more integrated. The thing here is frugivores behaviour is very relevant as they determine deposition sites of seeds that will later condition germination and seedling survival probabilities given that microhabitat suitability is very limited in arid environments.
Thus, the deposition site is relevant for seeds recruitment because not all microhabitats (i.e. the underneath or under canopy of different species) are equally suitable. I recommend try to make this idea more integrate and better explained.


L 70-71. The terms scarification and dehnibition should be better explained and related to how frugivores tratis, such us body size, via retention time.

L 74-75. "differences in effectiveness among dispersers are not only related to morphological tratis but also due to deposition patterns and microhabitat suitability".

L 183: Frequency of visitis: You sampled frequeny of visitis in two years periods but there is not sufficient detail about which trees were sampled in each year. Did you sample 7 trees in each succesional stage each year or 7 threes accounting for the two years?
In the case the secound is true you should better explain how many trees were sampled in each year in each succesional stage.

L 190: visitor species seems confusing, at least to me... I would recommend frugivore species
L 205 The hours reported don't seem to make sense to me. Nothing is specified about the number of mist nets used in each succesional stage, thus I understand is one mist-net in each succesional stage. In this case 1 mist-net * 15 days of sampling *11 hours of sampling yield 165 hours (15*11=165). In addition the size of the mist-net should be reported.

L 210. But I still do not understand how do you know when the bird consumed the fruit and when regurgitated or defectaed. If I understood well if seeds were provided at libitum birds could consume them any moment and thereby you cannot establish the moment of consumption and consequently neither seed retention time.

L 218: How did you establish body mass for birds? It was for each individual or for each species? Please provide this information in the manuscript.

L 251-252: If I understoond well, authors want to say here they measured birds body length and evaluated if such trait is related to the germination percentage of seeds defecated by each bird species. However this is not well expressed. Please rephrase it and additional details are needed. Did you measure captured birds? If this is the case I understand you have one data of length for each bird. The method and equipment used to measure the birds should be provided.

L 255: Subtitle "Seed deposition in secure sites (nurse plants)" I suggest to change secure sites (nurse plants) by potential suitable sites for recruitment.
I think I know understood what authors want to do by including the term of "secure site" or "nurse plant" I think you think this is important because due to the dry conditions in the open, seeds are probably not to be able to recruit outside the canopy of other plant species. If this is the case, this should be explained and supported with literature.

L 267-269 make no sense there...

L 178 - 277: replace "safe places" by "potential suitable microhabitats" I think this term is much more accurate to what authors mean.

L 216 Replace "vomit" by "regurgitation"

L 231: Delete "Meanwhile"
L 250: Poor English, difficult to understand. I suggest: "To evaluate the effecto f gut passage time duration on germination percentage.
L 251-251 Poor English, rephrase please.

L391: I would say that should be either complementary or redundacy, according to what is said before I would said that early-succesional stages have less redundancy in their dispersal community as consequence bird species result complementary and the services provided by each species are very valuable and likely irreplaceable in the early succesional stages.

L 480: I think there is a mistake here. The sentence doesn’t make sense.

L 490: Because B. longipes crown does not seem to be suitable for recruitment of conspecific seeds. I think it is interesting to add this information here.

L491-496: Did also more seeds arrive in M polyantha than other microhabitats? You could discuss here about what is being more limiting for B. longipes recruitment seed dispersal quantity or the availability of suitable microhabitats for recruitment.

L 519: Rephrase, poor English

L 520: Delete as a result of climate change (is repetitive with other part of the sentence)

L 524: "Change" repetition in the same line, please rephrase.

Experimental design

L 277: This is my strongest concern with the MS: Why do you "a priori" establish that Fabaceae species will be "safe places" (Weird name by the way "safe places"). I think that if authors want to "a priori" establish that Fabaceae species are more suitable microhabitats for B longipes recruitment than other plants they should provide further information supporting that. Otherwise you can make this part of the study (i.e. to evaluate microhabitat suitability among those available comparing Fabaceae with other types of plants).

Validity of the findings

I’m not very convinced about how gut passage time have been measured, in fact, the authors recognized this is more an estimation than an accurate measurement. Up to some point I don’t understand very well why statistical analyses are performed with such variable then… This being said, I don’t think this a big deal as the main findings of the article are not related to this result. Maybe when the results about gut passage time are discussed an advice that these results should be taken with caution may be OK.

Additional comments

I'm happy to say that the manuscript has much improved and that with few minor changes should be acceptable for publication. I have already highlighted some areas of poor English but I think a comprehensive English language review by an English native speaker should be done before the manuscript is accepted for publication. Here I provide a few suggestions and comments that I hope the authors find useful

---

## Round 0.3 · accepted · Accept

The authors have addressed satisfactorily all the comments and suggestions given by the reviewers.